# Prevalence of Steinert’s Myotonic Dystrophy and Utilization of Healthcare Services: A Population-Based Cross-Sectional Study

**DOI:** 10.3390/healthcare12080838

**Published:** 2024-04-16

**Authors:** Leticia Hernáez, Ana Clara Zoni, María-Felicitas Domínguez-Berjón, María D. Esteban-Vasallo, Cristina Domínguez-González, Pilar Serrano

**Affiliations:** 1Nursing Department, Faculty of Medicine, Universidad Autónoma de Madrid, 28040 Madrid, Spain; leticia.hernaez@estudiante.uam.es (L.H.); pilar.serrano@uam.es (P.S.); 2Subdirección General de Información Sanitaria, Ministry of Health, 28071 Madrid, Spain; 3Technical Unit for Health Status Report and Registries, Subdirección General de Vigilancia en Salud Pública, Dirección General de Salud Pública, 28040 Madrid, Spain; felicitas.dominguez@salud.madrid.org (M.-F.D.-B.); maria.estebanv@salud.madrid.org (M.D.E.-V.); 4Department of Neurology, Research Institute (imas12), Hospital Universitario 12 de Octubre, 28041 Madrid, Spain; cdgonzalez@salud.madrid.org; 5Instituto de Investigación Sanitaria Puerta de Hierro-Segovia de Arana (IDIPHISA), 28222 Majadahonda, Spain; 6Instituro Interuniversitario “Investigación Avanzada sobre Evaluación de la Ciencia y la Universidad” (INAECU), 28903 Madrid, Spain; 7Grupo de Investigación UAM “Vulnerabilidad Social, Cuidados y Salud” (GIVulneSCare), 28040 Madrid, Spain

**Keywords:** myotonic dystrophy, socioeconomic factors, health services administration

## Abstract

Myotonic dystrophy type I (MDI) is the most common muscular dystrophy in adults. The main objectives of this study were to determine the prevalence of MDI in the Community of Madrid (CM) (Spain) and to analyze the use of public healthcare services; a population-based cross-sectional descriptive study was carried out on patients with MDI in CM and data were obtained from a population-based registry (2010–2017). A total of 1101 patients were studied (49.1% women) with average age of 47.8 years; the prevalence of MDI was 14.4/100,000 inhabitants. In the women lineal regression model for hospital admissions, being in the fourth quartile of the deprivation index, was a risk factor (regression coef (rc): 0.80; 95%CI 0.25–1.37). In the overall multiple lineal regression model for primary health care (PHC) attendance, being a woman increased the probability of having a higher number of consultations (rc: 3.99; 95%CI: 3.95–5.04), as did being in the fourth quartile of the deprivation index (rc: 2.10; 95%CI: 0.58–3.63); having received influenza vaccines was a protective factor (rc: −0.46; 95%CI: −0.66–(−0.25)). The prevalence of MDI in the CM is high compared to other settings. Moreover, having any level of risk stratification of becoming ill (high, medium or low) has a positive association with increased PHC consultations and hospital admissions.

## 1. Introduction

According to the European Union’s criteria, rare diseases (RD) are those with a prevalence below 5 cases per 10,000 inhabitants. In Spain, an estimated 3 million people are affected by various RDs. Depending on the classification used, there are between 6000 and 8000 different entities, 80% are of genetic origin, and the disorder can be physical, sensory and/or mental, causing disability in 1 out of 3 patients [1].

Most RDs have been poorly studied and this limited knowledge hinders or delays diagnosis, making it impossible to take preventive or therapeutic measures. In turn, they are difficult to identify using the usual health information systems [2]. In 2015 the Rare Diseases Information System of the Community of Madrid (SIERMA) was created, a population-based registry that meets the information needs and serves as a basis for analyzing and studying the RDs treated in the Community of Madrid (CM) from an epidemiological perspective [3].

A large proportion of RDs are chronic diseases that cause high morbidity and premature mortality as well as a high degree of disability and, therefore, a significant decline in the quality of life of those affected [4]. Various studies have tried to establish the social and health impacts of patients affected by RDs. In 2012, a study estimated the financial burden on patients affected by different pathologies, with those with muscular dystrophy (Duchenne MD) bearing the greatest financial burden, including the informal costs that arise from care, using services and drugs. This study also analyzed quality of life for each disease, with muscular dystrophy again resulting in the poorest quality of life and the highest degree of dependence [5].

Steinert’s myotonic dystrophy or myotonic dystrophy type I (MDI) is the most common form of adult muscular dystrophy (disease classification code: OMIM 160900), with a total prevalence estimated at 12.5/100,000 inhabitants [6]. It is a rare genetic disease caused by a trinucleotide (CTG) expansion in the 3’UTRon chromosome 19q13.3 (DMPK gene)—a mutation in the 3 region on chromosome 19—which causes abnormalities in several systems, such as the muscular, respiratory, cardiac, endocrine, ocular, and central nervous systems [7]. Among other symptoms, it is characterized by progressive weakness (involving face, cervical, distal, and proximal muscles), myotonic phenomena, eyelid ptosis, cataracts, cardiac conduction disorders, sleep-disordered breathing and dysphagia [8].

The clinical variability of the manifestations of MDI requires a deep understanding and personalized treatment of each patient. Therefore, in 2018, a group of MDI experts developed a guide for the diagnosis and follow-up of this disease, identifying the professionals involved in MDI care, their activities, and establishing interdisciplinary collaboration [9]. This guide was created by several Spanish scientific societies.

Due to the large number of diseases and the great variability of their symptoms, it is difficult to reach a consensus on the main problems faced, but the literature shows that the lack of knowledge on the part of healthcare personnel, poor planning of healthcare services, and the difficulty of access for patients are the greatest difficulties encountered [10]. Patients with MDI have a high risk of suffering cardiac alterations—more specifically, rhythm alterations—which in most cases require hospital admission for pacemaker implantation [11,12]. Another of the major complications of this disease is pneumonia; these infections are caused by weakness of the respiratory muscles, which in most cases leads to hospital admission to treat the infection [13]. In addition to these problems, patients have an increased risk of cancer; there is no specific type of cancer, but they often have to be admitted for tests and treatment [14].

Because of these comorbidities, it is important to know what measures are taken to prevent them, including both influenza and pneumococcal vaccines, both of which have been shown to be effective in both the general population and in patients with neuromuscular disorders, significantly reducing admissions and complications from respiratory problems [15].

There are very few studies on the deprivation index and its relationship with RDs, although there is a growing body of work examining deprivation indices in different regions in order to analyze relationships between socioeconomic status, place of residence, and social health inequities such that people living in the most deprived areas have more health problems. It is essential to include an analysis of the distribution of health problems according to socioeconomic status in order to establish the necessary preventive and health promotion measures for the most vulnerable groups [16].

Currently, due to the complexity and variability of symptoms and the small number of patients affected by MDI, the studies available to us are limited to analyses of the clinical picture without delving into the factors involved in the multimorbidity that causes this RD, or the main reasons for hospitalization. It is therefore necessary to investigate the characteristics of the patients who are admitted and how these hospital admissions are, as well as their use of primary care, in order to improve the care provided to them by health service professionals. Therefore, the objectives of this study were as follows: to determine the prevalence of MDI in the Community of Madrid (CM) (Spain); to outline the sociodemographic profile of these patients; and to analyze the use of specialized care services, in addition to attendance at Primary Health Care (PHC), by these patients, taking into account sociodemographic factors and vaccination coverage. The results of this study may enable improvements to the care of these patients, as the knowledge gained will make it possible to implement the changes that these patients need in health services.

## 2. Materials and Methods

A descriptive cross-sectional study was carried out in the CM with a population of 6,542,630 inhabitants according to the census as of 31 July 2017 [17]. The study population was composed entirely of all CM residents with a health card during the study period (2010–2017) and a diagnosis of MDI. The source of information used was SIERMA (Rare Diseases Information System of the Community of Madrid), which compiles cases of MDI from the Hospital Discharges Data Set, Primary Care Electronic Health Record, Mortality Register, and patient registry of the RD Spanish Research Institute and from tertiary and referral hospitals for MDI in the CM [3]. To ensure validity, all cases were confirmed by medical record reviews prior to inclusion in the SIERMA population-based registry and were recorded with ORPHA 273 and ICD-10-ES G71.11 codes ORPHA 273, ICD-10-ES G71.11. Information on the use of health services was obtained from the Hospital Discharges Data Set and Primary Care Electronic Health Record. Data on influenza and pneumococcal vaccines were obtained from the Public Health Information System (SISPAL). In terms of reliability, standardized and automated processes were carried out for data extraction from the information systems used. 

The sociodemographic variables analyzed were: sex (woman/man); age at the end of the study period or age at death; and deprivation index by quartiles (based on the analysis of six indicators: manual worker population, casual worker population, unemployment, people aged 16 and over and 16–29 with a low education, and main dwellings without internet access), calculated for each basic health area of the PHC services assigned to the participants in the study (the first quartile indicates the best socio-economic status and the fourth quartile, the worst) [18,19]. Clinical and healthcare variables were studied for the 2010–2017 period; The coding of the variables is shown in Table 1. It should be noted that “level of risk stratification of becoming ill” (LRBI) (no risk, low risk, medium risk, high risk) predicted the needs of people who already have a chronic disease, based on the Kaiser pyramid, which—after analyzing 66 elements in the hospital and PHC setting—identifies three risk levels, with the ultimate aim of establishing the appropriate level of intervention for each risk level [20].

In the data analysis, we estimated MDI prevalence in 2017 with 95% confidence interval (95%CI), using the census from 1 July 2017 as the denominator. A descriptive analysis was carried out for all the study variables, calculating measures of central tendency and dispersion for quantitative variables and absolute frequencies and percentages for categorical variables. A bivariate analysis was performed between the dependent variables (hospital admissions, categorized as 0-1-2 and more admissions; and attendance in PHC, categorized from 0 to 10, from 11 to 20, and 21 or more consultations) and the rest of the sociodemographic and clinical variables, considered to be independent. Hypothesis testing was used according to the nature and distribution of the variables (Chi-squared, Student’s *t*-test, Mann–Whitney U test, Kruskal–Wallis test, ANOVA). A multivariate analysis was performed using multiple linear regression models for both the number of hospital admissions and the number of consultations in PHC. Total and sex-disaggregated analyses were carried out. The 95%CIs and *p*-value were calculated to determine whether statistical significance was less than or equal to 0.05. The STATA 18.0 statistical package was used. 

This study was conducted with the approval of the Regional Drug Research Ethics Committee of the Community of Madrid (approved by the Regional Drug Research Ethics Committee of the Community of Madrid with code DM1-CM on 17 December 2018). The confidentiality of the information was safeguarded in accordance with current Spanish data protection legislation and SIERMA legislation.

## 3. Results

A total of 1101 patients were studied in the 2010–2017 period, of whom 541 (49.1%) were female. The average age was 47.8 (standard deviation, SD: 17.03). By age group, 23.6% (n = 260) were in the 40–49 group and 23.8% (n = 262) were in the 18–39 group. Related to socioeconomic status, 30.4% (n = 335) of the patients were in the third quartile of the deprivation index with no statistically significant differences between quartiles. The average number of hospital admissions was 1.8 (SD: 3.79) and the average number of consultations in PHC was 13.5 (SD: 9.71), with statistically significant differences by sex (*p <* 0.001). People had a normal risk stratification score in the 48.6% (n = 535) with statistically significant differences by sex (*p* = 0.002) (Table 2).

The overall prevalence in 2017 was 14.44 cases/100,000 population (95%CI 13.55–15.39). The prevalence by sex was 14.28 (95%CI 13.06–15.60) in women and 14.62 (95%CI 13.35–16.03) in men (Table 3). 

The comparative analysis between people who had been admitted to hospital and those who had not showed statistically significant differences in all variables analyzed except for sex (*p* = 0.735). 44.4% (n = 489) of patients with no admissions. The average age of people who had no admissions was 43.9 (SD: 16.42) and as the average age increased, the number of admissions increased (*p <* 0.001). Of the people who had no admissions, 62.6% (n = 306) were under 50 years old compared to people with two or more admissions, 58.6% (n = 222) of whom were 50 years old or above (*p <* 0.001); 31.7% (n = 120) of patients with two or more admissions were in the third quartile of the deprivation index, and 28.3% (n = 66) of those with one admission were in the fourth quartile (*p* = 0.003). The number of influenza vaccinations received during the 2010–2017 period revealed differences between individuals with no admissions and those with admissions, with the number of doses decreasing among those who had been admitted (*p <* 0.001). Additionally, 72.1% (n = 168) of people with an admission had not received the pneumococcal vaccine (*p <* 0.001), 45.8% (n = 224) of people who had no admissions had a normal risk level, and 4.8% (n = 18) of people who had two or more admissions had a high-risk stratification score (*p <* 0.001) (Table 4). 

A total of 1940 hospital admissions were analyzed, yielding an average age of 49.7 (SD: 20.1), with 55.1% (n = 1068) of admissions occurring in men. The median length of stay for single admissions was 1 (0–410). By type of admission, 47.6% (n = 923) were emergency admissions, and the Internal Medicine department accounted for the largest number of admissions, with 396 (20.4%) of the total. The main reasons for admissions were respiratory diseases (21.4%), cardiovascular diseases (13.1%), and neoplasms (9.8%) (Table 5).

In total, 45.9% (n = 505) of the people visited PHC between 0 and 10 times, with differences in all variables except age groups. The average age of these patients was 47.9 (SD: 15.91), and the average number of influenza vaccines they had received was 6.3 (SD: 2.55). The 67.6% (n = 146) of those who visited more than 20 times were women, compared to 41% (n = 207) of those who visited between 0 and 10 times (*p <* 0.001). In the third and fourth quartiles of the deprivation index were the 31.9% (n = 161) of patients who visited PHC between 0 and 10 times and the 31.5% (n = 68) of those who visited more than 20 times, respectively. The 60.6% of patients who visited between 0 and 10 times had a normal LRBI (*p <* 0.001) (Table 6). 

In the overall linear regression model for hospital admissions, having a low LRBI was a protective factor (regression coefficient, rc: −0.71; 95%CI −1.22–(−0.19)). The number of influenza vaccinations received also had a protective effect (rc: −0.18; 95%CI −0.27–(−0.09)), while pneumococcal vaccination was a risk factor (rc: 0.84; 95%CI 0.28–1.39). In the women model, being in the fourth quartile of the deprivation index, the most disadvantaged, was a risk factor (rc: 0.80; 95%CI 0.25–1.37) compared to the men model, where no statistically significant differences were found. In both the men and women models, the number of influenza vaccinations acted as a protective factor and in the men model, pneumococcal vaccination had a risk effect (rc: 1.30; 95%CI 0.35–2.24) (Table 7).

In the overall multiple linear regression model for PHC attendance, being a woman increased the probability of having a higher number of consultations (rc: 3.99; 95%CI: 3.95–5.04), while belonging to the fourth quartile of the deprivation index was associated with a higher risk (rc: 2.10; 95%CI: 0.58–3.63). Having received influenza vaccines was a protective factor (rc: −0.46; 95%CI: −0.66–(−0.25)), while having the pneumococcal vaccine was a risk factor (rc: 1.50; 95%CI: 0.5–3.12). Hospital admissions increased the likelihood of consultations (rc: 2.87; 95%CI: 1.77–3.96). Having an LRBI was a risk factor (rc: 7.03; 95%CI: 5.46–8.60). In the women model, being in the fourth quartile of the deprivation index was a risk factor (rc: 2.80; 95%CI 0.21–5.39), while in men, it was not statistically significant. Having received the pneumococcal vaccine increased the risk in the men model (rc: 2.49; 95%CI 1.06–3.92). However, in women, it was not statistically significant (Table 8). 

## 4. Discussion

There are currently no known epidemiological studies on MDI, so this is ground-breaking research on the epidemiology of this health problem, focusing primarily on the use of healthcare services.

The first notable finding is that in the Community of Madrid, a higher prevalence has been recorded than that reported in the literature [6], with no differences by sex being observed, although there are differences by age group. In the Spanish region of Guipúzcoa, an epidemiological study on MDI was carried out in 1993, resulting in a prevalence of 26.5 per 100,000 inhabitants, much higher than that analyzed in the literature [21]; another more recent study conducted in a region of northern Spain analyzed the prevalence of the different muscular dystrophies, resulting in a prevalence of 35.9/100,000 inhabitants for MDI [22]. This may be due to the fact that population-based registries, such as SIERMA, are extremely comprehensive when it comes to collecting data, drawing on a number of different registers, so they provide a more accurate picture of prevalence [23]. According to Orphanet (a portal dedicated to the study of RDs), the prevalence of MDI stands at 12.5 cases/100,000 inhabitants [24]. The prevalence of other degenerative neuromuscular diseases with similar characteristics is far lower, e.g., amyotrophic lateral sclerosis, with a prevalence of 3–6/100,000 people [25], Duchenne muscular dystrophy with 5/100,000 [26], and other muscular dystrophies such as limb-girdle or facioscapulohumeral muscular dystrophy [26].

The literature shows different classifications of MDI based on age at onset, but there is a consensus that muscular dystrophy is more common in adults [27]. Most symptoms begin in adulthood—more specifically in the third and fourth stages of life [28]—which is consistent with our study, where we found an average age of 48 years. The percentage of men and women is also very similar to other studies, with an average age of 47.3 in a study carried out in Japan in 2022 with a 53.2% women sample. Another study, also conducted in Japan in 2020, found 49% women and an average age of 47 [29]. 

Our findings reflect that less than one third of the patients received the pneumococcal vaccine and there were more admissions among patients who had received the pneumococcal vaccine, despite the literature indicating the need for vaccination in at-risk patients [30]. This could be explained by the fact that health professionals in the hospital setting, having more contact with these patients, are more insistent on the need for the pneumococcal vaccine. 

Although annual influenza vaccination coverage was low in the present study, it did lead to a decrease in admissions and consultations in PHC. Despite the fact that it is generally recommended to vaccinate all patients with neuromuscular diseases [15], in 2017, only 30% of patients received the influenza vaccine, which means that the coverage of these patients is far below that of the general population. According to vaccination coverage data in the CM, during 2010–2017, coverage was close to 57% for the population over 65 years of age and was 30% for the population aged 60–64, so we do not have data for the younger population, which makes up the largest percentage of our population [31]. There are numerous studies linking influenza vaccination among patients with chronic diseases to a significant reduction in admissions [32]. Studies have shown the effectiveness of the influenza vaccine in reducing recurrent hospitalizations among patients with coronary heart disease and respiratory problems by up to 6% [33]. Other studies suggest that it may reduce occurrence of cardiovascular diseases [34].

MDI is a disease that is very often associated with respiratory complications, which are one of the main causes of death and hospital admission [35]; this is consistent with our findings, where the primary diagnoses were respiratory diseases (21.4%). A study carried out in the Community of Madrid on Angelman syndrome, an RD similar to MDI, where the causes of admission were analyzed using the Minimum Basic Data Set [36], showed that these patients also have a high number of admissions for respiratory problems. Furthermore, in our study, 13% of admissions were due to cardiovascular problems, which is consistent with the literature, which shows that a high number of patients suffer from arrhythmias and other cardiac rhythm disturbances [11]. It should be noted that there are numerous studies showing a higher incidence of cancer in patients with MDI, which is in line with our study where almost 10% of admissions were due to oncological problems [37]. 

Our study shows that patients in the most deprived quartiles of the deprivation index attended more primary care and also had more hospital admissions in the case of women. As already reflected in the literature, low socioeconomic status is one of the key structural social determinants of health inequalities [38]; epilepsy incidence increased in areas with a lower deprivation index [39], as did cancer mortality [40], anxiety in cancer patients [41], and poorer health and quality of life [42]. There are currently no studies available on the frequency of use of PHC for RDs nor on the profile of patients admitted for RDs, but we can find studies on the use of services in the general population according to socioeconomic status. In Catalonia, it was shown that users with a lower socioeconomic status required more time in consultations [43]. Another study conducted in Spain, where attendance was analyzed in the general population, showed that, as in our study, women were more likely to attend consultations and that belonging to a lower social class also increased the likelihood of this [44]. 

Regarding the level of risk stratification, 3.6% were in the high-risk group, a figure very similar to that found by Barrio-Cortés et al. [45], and we also found that any level of risk was associated with an increase in the number of consultations in primary care, results similar to those found in other studies in our setting [46,47].

Gender inequalities have been widely demonstrated, and in recent years, attempts have been made to analyze these inequalities in order to provide tools to narrow the gap. In our study, being female increased the likelihood of attending PHC consultations, which is consistent with the literature. A Europe-wide study analyzing databases from different countries over 15 years concluded that ill health was 17% more likely in women than in men and was especially higher in women with a lower level of education [48]. In Spain, a study was conducted in the 2006–2017 period to analyze gender inequalities in access to PHC and hospital emergency services, finding a higher use of PHC services among women than men, and even more so among those with a lower level of education [49]. 

### 4.1. Limitations

The main limitation of this study is related to use of documentary sources, which only provide basic sociodemographic data. We have therefore not been able to take into account other relevant variables such as educational level, occupation, income level, and migration status, which are clear axes of inequality and which allow us to understand the dynamics of health problems [50].

In addition, the existence of other clinical variables, including data such as the reasons for consultation in PHC and the type of professional who carried out the consultation, could expand the information needed to understand the use of health services in greater depth, but it was not possible to examine this because these variables were not found in the documentary sources used either.

On the other hand, the cross-sectional nature of the study limits the possibility of establishing causal relationships.

### 4.2. Theoretical and Practical Implications 

This study opens up lines of research, especially regarding the use of services, as there are currently limited studies in this field and the few that exist are focused on motor diseases that are not the subject of our study. Future longitudinal studies with primary data are needed, which—in addition to incorporating socioeconomic variables—would allow us to monitor the clinical evolution of these patients with MDI.

MDI can be considered a chronic disease due to the pathologies that derive from it (especially cardiac and respiratory), and therefore, risk stratification scores can be a very useful indicator for measuring the impact of morbidity and are necessary for good health planning [51]. Despite this, it is not easy to locate studies where risk stratification is used as a variable, so it would be necessary to implement it in a generalized way in clinical history and to take it into consideration for future studies.

In our study, cardiovascular and respiratory problems were among the main reasons for hospital admissions, which could be reduced with an increase in vaccination coverage according to the literature, both for influenza and pneumococcus; however, it is important to increase studies on this issue, as the data obtained have not provided a complete understanding of the influence of vaccines, especially pneumococcal vaccination.

All the aforementioned complications should be monitored in PHC consultations, as it is a progressive disease and ongoing check-ups can help prevent complications, palliate symptoms, and improve the quality of life of these patients [35]. There are few studies on the impact of PHC on the progression of the disease, but we did find some in which it is regarded as key to both the diagnosis and treatment of other neuromuscular diseases [31,52].

## 5. Conclusions

The prevalence of MDI in the Community of Madrid is high compared to other settings. There were no significant differences between sex, but there were significant differences by age group, with a higher prevalence among people over the age of 18. People who have received a higher number of doses of influenza vaccine are less likely to be admitted to hospital and less likely to require PHC consultations. Women are more likely to attend PHC consultations. Moreover, having any level of risk stratification, whether high, medium, or low, is a risk factor for a higher number of PHC consultations and hospital admissions.

## Figures and Tables

**Table 1 healthcare-12-00838-t001:** Coding of clinical and healthcare variables.

Clinical and Healthcare Variables	Items
Number of influenza vaccines administered	0 1–7 8 vaccines
Pneumococcal vaccine	Yes/No
Hospital Admissions	Numeric
Days of hospital stay	Numeric
Type of admission	Urgent/scheduled
Admitting department	Internal Medicine Pneumology Cardiology Digestive Oncology Other specialties No record
Primary diagnosis	Respiratory diseases Cardiovascular diseases Neoplasms Digestive disorders Diseases of the central nervous system Other pathologies
Attendance in PHC	Numeric
Any level of risk stratification of becoming ill (LRBI)	No risk Low risk Medium risk High risk

**Table 2 healthcare-12-00838-t002:** Description of patients with Steinert’s myotonic dystrophy. Total and disaggregated by sex. Period 2010–2017.

	Total n = 1101		Women n = 541 (49.1%)		Men n = 560 (50.9%)		*p* Value
	Mean (SD)	Median (Min–Max)	Mean (SD)	Median (Min–Max)	Mean (SD)	Median (Min–Max)	
Age	47.8 (17.03)	48 (0–98)	48.8 (16.02)	49 (2–98)	47 (17.92)	47 (0–94)	0.170
Influenza vaccine	5.7 (2.84)	7 (0–8)	5.7 (2.89)	7 (0–8)	5.8 (2.8)	7 (0–8)	0.952
Consultations in primary care	13.5 (9.71)	11 (0–69)	15.7 (10.95)	14 (0–69)	11.4 (7.79)	10 (0–59)	<0.001
Hospital admissions	1.8 (3.79)	1 (0–71)	1.6 (2.3)	1 (0–15)	1.9 (4.81)	1 (0–71)	0.631
	n (%)		n (%)		n (%)		
Age group							0.275
<18	54 (4.9)		19 (3.5)		35 (6.3)		
18–39	262 (23.8)		130 (24)		132 (23.6)		
40–49	260 (23.6)		126 (23.3)		134 (24)		
50–59	239 (21.7)		128 (23.7)		111 (19.8)		
60–69	188 (17.1)		92 (17)		96 (17.1)		
>70	98 (8.9)		46 (8.5)		52 (9.3)		
Deprivation index							0.583
1st quartile (less deprived)	237 (21.5)		107 (19.8)		130 (23.2)		
2nd quartile	247 (22.4)		123 (22.7)		124 (22.1)		
3rd quartile	335 (30.4)		169 (31.2)		166 (29.6)		
4th quartile (most deprived)	282 (25.6)		142 (26.3)		140 (25)		
Risk level							0.002
Normal	535 (48.6)		233 (43.1)		302 (53.9)		
Low risk	362 (32.9)		202 (37.3)		160 (28.6)		
Medium risk	163 (14.8)		89 (16.5)		74 (13.2)		
High risk	40 (3.6)		16 (3)		24 (4.3)		
Death							
Yes	156 (14.2)		54 (10)		102 (18.2)		<0.001
No	945 (85.8)		487 (90)		458 (81.8)		
Influenza vaccine 2017							0.119
Yes	340 (30.9)		179 (33.1)		161 (28.8)		
No	761 (69.1)		362 (66.9)		399 (61.3)		
Influenza vaccine							0.730
0	82 (7.5)		42 (7.8)		40 (7.1)		
1–7 vaccines	485 (44.1)		232 (42.9)		253 (45.2)		
8	534 (48.5)		267 (49.4)		267 (47.7)		
Pneumococcal vaccine							0.256
Yes	331 (30.1)		154 (24.5)		177 (31.6)		
No	770 (69.9)		387 (71.5)		383 (68.4)		
Hospital admissions							
0	489 (44.4)		234 (43.3)		255 (45.6)		0.862
1	233 (21.2)		118 (21.8)		115 (20.5)		
2	127 (11.5)		65 (12)		62 (11.1)		
3	85 (7.7)		45 (8.3)		40 (7.1)		
4	55 (5)		24 (4.4)		31 (5.5)		
5+	112 (10.2)		55 (10.2)		57 (10.2)		
Consultations in PHC							<0.001
0–5	215 (19.5)		80 (14.8)		135 (24.1)		
6–10	290 (26.3)		127 (23.5)		163 (29.1)		
11–15	230 (20.9)		107 (19.8)		123 (22)		
16–20	150 (13.6)		81 (15)		69 (12.3)		
20+	216 (19.6)		146 (27)		70 (12.5)		

SD: Standard deviation.

**Table 3 healthcare-12-00838-t003:** Prevalence of Steinert’s myotonic dystrophy. Total and disaggregated by sex. Calculated per 100,000 inhabitants. Year 2017.

	Women	95%CI	Men	95%CI	Total	95%CI
≤18	3.56	2.35–5.39	6.46	4.78–8.73	5.05	3.95–6.44
>19	16.65	15.2–18.23	16.76	15.23–18.45	16.7	15.63–17.84
Total	14.28	13.06–15.30	14.62	13.35–16.03	14.44	13.55–15.39

95%CI: 95% confidence interval.

**Table 4 healthcare-12-00838-t004:** Frequency of hospital admissions in patients with Steinert’s myotonic dystrophy according to clinical and sociodemographic variables. Period 2010–2017.

	0 Admissions n = 489 (44.4%)			1 Admission n = 233 (21.2%)			2 or More Admissions n = 379 (34.4%)			*p* Value
	Mean (SD)	95%CI	Median (Min–Max)	Mean (SD)	95%CI	Median (Min–Max)	Mean (SD)	95%CI	Median (Min–Max)	
Age	43.9 (16.42)	42.42–45.34	44 (0–98)	50.6 (14.64)	48.69–52.46	50 (2–92)	51.3 (18.10)	49.44–53.09	53 (2–92)	<0.001
Influenza vaccine	6.3 (2.61)	6.11–6.57	8 (0–8)	5.6 (2.92)	5.25–6	7 (0–8)	5 (2.91)	4.68–5.26	6 (0–8)	<0.001
Days of hospital stay				5.9 (11.92)	4.41–7.48	2 (0–92)	26 (39.83)	21.84–30.12	14 (0–433)	<0.001
	n (%)			n (%)			n (%)			
Sex										
Men	255 (52.2)			115 (49.4)			190 (50.1)			0.735
Women	234 (47.9)			118 (50.6)			189 (49.9)			
Age group										
<50 years	306 (62.6)			113 (48.5)			157 (41.4)			<0.001
50 or more	183 (37.4)			120 (51.5)			222 (58.6)			
Deprivation index										
1st quartile	127 (26)			43 (18.5)			67 (17.7)			0.003
2nd quartile	103 (21.1)			64 (27.5)			80 (21.1)			
3rd quartile	155 (31.7)			60 (25.8)			120 (31.7)			
4th quartile	104 (21.3)			66 (28.3)			112 (29.6)			
Risk level										
Normal	224 (45.8)			105 (45.1)			206 (54.4)			<0.001
Low risk	205 (41.9)			78 (33.5)			79 (20.8)			
Medium risk	49 (10)			38 (16.3)			76 (20.1)			
High risk	10 (2)			12 (5.2)			18 (4.8)			
Influenza vaccination										
0	31 (6.3)			18 (7.7)			33 (8.7)			<0.001
1–7 vaccines	160 (32.7)			102 (43.8)			223 (58.8)			
8	298 (60.9)			113 (48.5)			123 (32.5)			
Pneumococcal vaccination										
Yes	97 (19.8)			65 (27.9)			169 (44.6)			<0.001
No	392 (80.2)			168 (72.1)			210 (55.4)			

SD: standard deviation; 95%CI: 95% confidence interval.

**Table 5 healthcare-12-00838-t005:** Descriptive of hospital admissions in patients with Steinert’s myotonic dystrophy (n = 1940). Period 2010–2017.

	Mean (SD)	Median (Min–Max)
**Days of hospital stay**	5.4 (15.15)	1 (0–410)
	n (%)	
**Type of admission**		
Day care	338 (17.4)	
Emergency	923 (47.6)	
Scheduled	679 (35)	
**Admitting department**		
Internal Medicine	396 (20.4)	
Pneumology	307 (15.8)	
Cardiology	195 (10.1)	
Digestive	164 (8.5)	
Oncology	159 (8.2)	
No record	91 (4.7)	
Other specialities	628 (32.4)	
**Primary diagnosis**		
Respiratory diseases	416 (21.4)	
Cardiovascular diseases	254 (13.1)	
Neoplasms	190 (9.8)	
Digestive disorders	188 (9.7)	
Diseases of central nervous system	175 (9)	
Other pathologies	717 (37)	

SD: standard deviation.

**Table 6 healthcare-12-00838-t006:** Attendance in PHC of patients with Steinert’s myotonic dystrophy according to clinical and sociodemographic variables. Period 2010–2017.

	0–10 Consultations n = 505 (45.9%)			11–20 Consultations n = 380 (34.5%)			21 or More Consultations n = 216 (19.6%)			*p* Value
	Mean (SD)	95%CI	Median (Min–Max)	Mean (SD)	95%CI	Median (Min–Max)	Mean (SD)	95%CI	Median (Min–Max)	
Age	47.9 (15.91)	46.55–49.33	49 (0–94)	48.6 (17.19)	46.83–50.29	49 (3–92)	46.3 (19.14)	43.78–48.89	47 (2–98)	0.3612
Influenza vaccination	6.3 (2.55)	6.04–6.49	8 (0–8)	5.5 (2.92)	5.16–5.75	7 (0–8)	4.9 (3.09)	4.49–5.32	6 (0–8)	0.0001
	n (%)			n (%)			n (%)			
Sex										
Men	298 (59)			192 (50.5)			70 (32.4)			<0.001
Women	207 (41)			188 (49.5)			146 (67.6)			
Age group										
<50 years	262 (51.9)			195 (51.3)			119 (55.1)			0.651
50 or more	243 (48.1)			185 (48.7)			97 (44.9)			
Deprivation index										
1st quartile	120 (23.8)			89 (23.4)			28 (13)			0.014
2nd quartile	102 (20.2)			89 (23.4)			56 (25.9)			
3rd quartile	161 (31.9)			110 (29)			64 (29.6)			
4th quartile	122 (24.2)			92 (24.2)			68 (31.5)			
Risk level										<0.001
Normal	306 (60.6)			159 (41.8)			70 (32.4)			
Low risk	156 (30.9)			135 (35.5)			71 (32.9)			
Medium risk	34 (6.7)			69 (18.2)			60 (27.8)			
High risk	9 (1.8)			17 (4.5)			14 (6.5)			
Influenza vaccination										
0	25 (5)			31 (8.2)			26 (12)			<0.001
1–7 vaccines	195 (38.6)			182 (47.9)			108 (50)			
8	285 (56.4)			167 (44)			82 (38)			
Pneumococcal vaccination										
Yes	125 (24.8)			119 (31.3)			87 (40.3)			<0.001
No	380 (75.3)			261 (68.7)			129 (59.7)			

SD: standard deviation; 95%CI: 95% confidence interval.

**Table 7 healthcare-12-00838-t007:** Multiple linear regression models for hospital admissions. Total and disaggregated by sex. Period 2010–2017.

		Total Model			Women Model			Men Model		
		rc	95%CI	*p* Value	rc	95%CI	*p* Value	rc	95%CI	*p* Value
Sex (women)		−0.21	−0.89	0.35						
Age		−0.007	−0.03	0.349	0.01	−0.023	0.14	−0.01	−0.05	0.27
Deprivation index	Quartile 2	0.11	−1.33	0.755	0.34	−1.16	0.246	−0.15	−2.36	0.802
	Quartile 3	0.2	−1.24	0.529	0.39	−1.09	0.162	0.15	−2.19	0.792
	Quartile 4	0.3	−1.29	0.368	0.8	0.25–1.37	0.005	−0.15	−2.28	0.794
Risk level	low	−0.71	−1.03	0.007	−0.67	−0.77	0.002	−0.7	−1.89	0.143
	medium	−0.12	−1.33	0.733	0.25	−1.1	0.369	−0.55	−2.46	0.38
	high	−0.7	−2.42	0.26	−0.07	−2.26	0.9	−1.14	−4.01	0.264
Number of influenza vaccines		−0.18	−0.18	<0.001	−0.14	−0.15	<0.001	−0.23	−0.32	0.004
Pneumococcal vaccine (received)		0.84	0.28–1.39	0.001	0.003	−0.98	0.303	1.3	0.35–2.24	0.007
Constant		3.1	1.98–4.2	0	1.47	0.57–2.37	0.001	3.82	1.86–5.79	0
Number of observations		Total model			Women model			Men model		
		1101			541			560		
F		5.14			5.79			3.13		
Prob > F		0			0			0.001		
R-squared		0.05			0.1			0.049		

rc: regression coefficient; 95%CI: 95% confidence interval.

**Table 8 healthcare-12-00838-t008:** Multiple linear regression models for PHC attendance. Total and disaggregated by sex.

		Total Model			Women Model			Men Model		
		rc	95%CI	*p* Value	rc	95%CI	*p* Value	rc	95%CI	*p* Value
Sex (women)		3.99	2.95–5.04	<0.001						
Age		−0.1	−0.06	<0.001	−0.14	−0.13	<0.001	−0.07	−0.07	<0.001
Deprivation index	Quartile 2	2.05	0.49–3.62	0.01	2.85	−5.69	0.036	1.35	−3.52	0.133
	Quartile 3	1.46	−2.911	0.05	1.67	−4.98	0.188	1.3	−3.27	0.118
	Quartile 4	2.1	0.58–3.63	0.007	2.8	0.21–5.39	0.034	1.53	−3.43	0.08
Risk level	Low	2.82	1.61–4.04	<0.001	3.91	0.93–4.90	0.004	2.79	1.37–4.21	<0.001
	Medium	7.03	5.46–8.60	<0.001	8.17	5.62–10.72	<0.001	5.56	3.72–7.39	<0.001
	High	5.9	3.06–8.74	<0.001	5.37	0.14–10.59	0.044	5.91	2.91–8.92	<0.001
Number of influenza vaccines		−0.46	−0.41	<0.001	−0.49	−0.69	0.005	−0.44	−0.48	<0.001
Pneumococcal vaccine (received)		1.5	0.5–3.12	0.007	0.72	−4.57	0.535	2.49	1.06–3.92	0.001
Admission (admitted)		2.87	1.77–3.96	<0.001	3.54	1.70–5.37	<0.001	2.04	0.79–3.30	0.001
Constant		13.18	10.54–15.83	<0.001	18.38	3.90–22.86	<0.001	14.28	9.30–15.26	<0.001
Number of observations		Total			Women model			Men model		
		1101			541			560		
F		23.04			9.24			12.75		
Prob > F		0			0			0		
R-squared		0.2026			0.1612			0.1884		

rc: regression coefficient; 95%CI: 95% confidence interval.

## Data Availability

Data are contained within the article.

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
