# Peer review of "Prevalence of Steinert’s Myotonic Dystrophy and Utilization of Healthcare Services: A Population-Based Cross-Sectional Study"

_healthcare, 2024, doi:10.3390/healthcare12080838_

Round 1

Reviewer 1 Report

Comments and Suggestions for Authors

In the abstract, the method and results should be stated in more detail. In abstract, Methods, study type, location, time and sample size and sampling method is needed.                                                                 

In abstract Results, a description of basic and demographic variable is needed.

Check the keywords based on the Mesh.

In the introduction, the gap should be stated.

The method should be revised based on STROBE checklist: cross sectional study.

Data collection tools and validity and reliability is required.

What’s Global mean??

In Table 1, what statistical test is used?

What is the basis of age division in Table 2?

There is no significance of 0.00. Be corrected to > 0.001.

Table 4 is unclear.

There are many tables and it is difficult to understand.

Has the admission trend changed?

In the discussion, all the sentences need a reference, some parts are long.

Reviewer 2 Report

Comments and Suggestions for Authors

Reviewer Report for Prevalence of Steinert´s myotonic dystrophy and utilization of 2 healthcare services: A population-based-cross-sectional study

Leticia Hernáez, Ana Clara Zoni , María Felicitas Domínguez, María Dolores Esteban, Cristina Domínguez , Pilar Serrano and on behalf of the DM-CM1 Working Group†

In this article the socio-demographic profile of patients with Steinert's Myotonic Dystrophy is discussed.

Abstract

·           Please observe the structure of the abstract: relevance, goal, materials and methods, results and discussion, conclusion.

·           Line 25. Move the phrase "the prevalence of DMI was 14.4/100,000 inhabitants" to the results.

·           Line 28 (Coef: 0.80; 95%CI 0.25–1.37). Note that this is a regression coefficient.

·           Line 29 and 33. Phrase “being a woman increased the probability of having a higher number of consultations” is duplicated by the phrase “Women are more likely to attend PHC consultations”.

Introduction

In this section, please refer to the literature on factors affecting hospitalization and attendance at PHC people are affected by variable RDs. Move the paragraphs of Lines 251-270, 284-300, 301-322 to this section. Explain why it is important to study this aspect? Explain, in this regard, what problem your study solves.

Results

Coefficient of determination (R-squared) in Tables 6 and 7 have very low values, which indicates a very weak association of the studied factors with the response. How much does linear regression models apply here? Your reasoning may be baseless. The relative risk and odds ratio count seems to be more informative.

Discussion

·       The section contains numerous data into the factors involved in the multimorbidity that causes this RD, or the main reasons for hospitalization. (Lines 251-270, 284-300, 301-322). I recommend moving this discussion to the Introduction section.

·       In the section, focus on your own original results, and less on previously obtained data.

Reviewer 3 Report

Comments and Suggestions for Authors

The topic seemed topical and exciting.

I have a few reservations about the paper:

1.      Line In the data analysis, we estimated DMI prevalence in 2017…

Why did the authors select the year 2017?

Are there any particular incidents happening at that point in time? OR

How do the authors justify their selection now?

The authors should provide a better justification for their data selection

2.      Literature Review is a section from which readers can better understand what has already been done in the area. In the current format, the authors directly provide the materials. Thus, readers could face difficulties in understanding how the variables are selected for the study. Thus, the authors should provide a vivid background of the study context.

3.      The authors mentioned several acronyms in their papers, such as ORPHA, ICD, etc.

I will suggest they provide a list mentioning the complete form of these acronyms.

4.      Line 105-106 The following clinical and healthcare variables… administered

The authors described the “Coding” process of several diseases in detail. It is pretty difficult for the readers to follow all the “coding.”

I will suggest a better option, like using a table, to the authors. It will help the authors to read and understand all “coding” in a better manner.

5.      Line 285 romuscular diseases [31], in 2017, only 30%...of patients received the

The authors can provide a reason why the coverage is far behind.

It could provide a better contribution to the current study. The authors can even mention any other incidents in other countries and compare the current situation with the previous one.

6.      Line 320 There are few studies on the impact of PHC… of the disease,

The authors mentioned that a few studies discussed the impact of PHC on the progression of the disease. However, they mention only only References. Thus, they must provide a few more references to justify their claim.

Secondly, how are their findings different from the current ones?

7.      The authors should talk about the paper's "Implications (Theoretical and Practical)" separately.

8.      The limitation section is a little bit confusing. According to the authors, a few more socio-demographic variables should be included to make the study more complete.

However, several studies used different methodologies to explore the findings. How is the current study better than the one?

I suggest the authors provide a better understanding of the limitations of the current study.

Comments on the Quality of English Language

Moderate English corrections make the paper more legitimate to the readers.
